# Neurosymbolic Reasoning Shortcuts under the Independence Assumption

**Emile van Krieken,**
Emile.van.Krieken@ed.ac.uk
*University of Edinburgh*

**Pasquale Minervini**[*],
*University of Edinburgh, Miniml.AI*

**Edoardo Ponti**[*]**, Antonio Vergari**[*]
*University of Edinburgh*

**Editors:** Leilani H. Gilpin, Eleonora Giunchiglia, Pascal Hitzler, and Emile van Krieken

## Abstract

The ubiquitous independence assumption among symbolic concepts in neurosymbolic (NeSy) predictors is a convenient simplification: NeSy predictors use it to speed up probabilistic reasoning. Recent works like van Krieken et al. (2024) and Marconato et al. (2024) argued that the independence assumption can hinder learning of NeSy predictors and, more crucially, prevent them from correctly modelling uncertainty. There is, however, scepticism in the NeSy community about the scenarios in which the independence assumption actually limits NeSy systems (Faronius and Dos Martires, 2025). In this work, we settle this question by formally showing that assuming independence among symbolic concepts entails that a model can never represent uncertainty over certain concept combinations. Thus, the model fails to be aware of *reasoning shortcuts*, i.e., the pathological behaviour of NeSy predictors that predict correct downstream tasks but for the wrong reasons.

## 1. Introduction

Neurosymbolic (NeSy) predictors are a class of models that combine neural perception with symbolic reasoning (Manhaeve et al., 2021; Xu et al., 2018; Badreddine et al., 2022; Feldstein et al., 2024; Ahmed et al., 2022; De Raedt et al., 2019; Hitzler et al., 2022; Garcez and Lamb, 2023) to obtain more transparent and reliable ML systems, especially for safety-critical applications (Giunchiglia et al., 2023). A common recipe is that of *probabilistic* NeSy predictors (Xu et al., 2018; Manhaeve et al., 2018; Ahmed et al., 2022; van Krieken et al., 2023). First, they use a neural network to extract probabilities for *concepts*, which are symbolic representations of the input. Then, it uses probabilistic reasoning (Darwiche and Marquis, 2002) over an interpretable symbolic program to predict the final labels. Used effectively, this can lead to interpretable and reliable AI systems.

When the data and program together do not constrain the neural network sufficiently, NeSy predictors can learn *reasoning shortcuts* (RSs) (Marconato et al., 2023). RSs are incorrect mappings from the input to concepts that are consistent with the program and the data. Unfortunately, when a NeSy predictor learns an RS, it will not generalise out-of-distribution, breaking the promised reliability of NeSy predictors. What should we do if we cannot avoid RSs? Marconato et al. (2024) argues to make our predictors *aware* of the presence of RSs. This comes down to representing uncertainty over all concept assignments consistent with the data. To highlight this idea, we consider the XOR MNIST task.

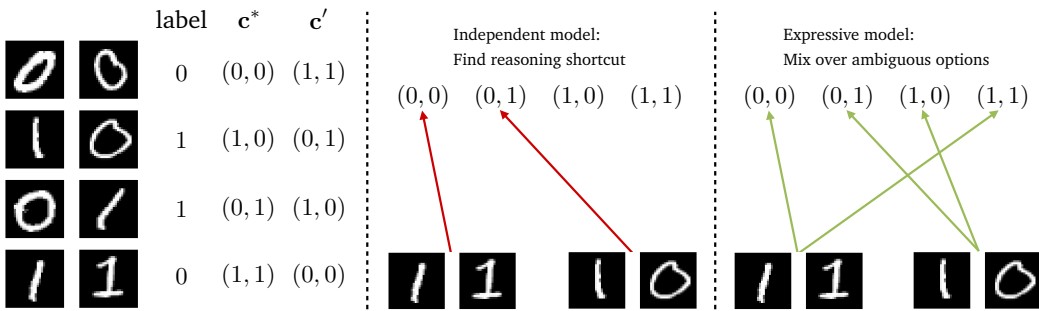

Figure 1: **Independent models can be overconfident in the presence of reasoning shortcuts.** Left: The XOR MNIST task contains a reasoning shortcut when we swap 0's and 1's. Middle: A conditionally independent model learns this reasoning shortcut with high probability, becoming overconfident in the wrong prediction. Right: An expressive model can express uncertainty over the correct prediction with the right loss function. **This is not possible for independent models.**

**Example 1** *In the XOR MNIST task (Fig. 1) we have pairs of MNIST images of 0 and 1. The goal is to model the XOR function: if the MNIST images represent different digits, the label is 1, and otherwise it is 0. Without direct feedback on the digits, we cannot disambiguate between mapping 0 to 0 and 1 to 1, or 0 to 1 and 1 to 0. Instead, an RS-aware model, given images 1 and 0, assigns 0.5 probability to the digit pairs consistent with the data, namely $(0, 1)$ and $(1, 0)$.*

Most applications of probabilistic NeSy predictors make a crucial simplifying assumption, namely that the concepts extracted by the neural network are *conditionally independent* (van Krieken et al., 2024). This assumption simplifies and speeds up a computation that is, in general, intractable (Chavira and Darwiche, 2008). Unfortunately, van Krieken et al. (2024) showed that the independence assumption results in loss functions with disconnected and non-convex loss minima, complicating training, and preventing the model from expressing dependencies between concepts. Furthermore, it can bias the solutions towards "determinism", i.e., solutions where only a few concepts receive all the probability mass. These results were recently questioned by Faronius and Dos Martires (2025), who argue that the setting investigated in van Krieken et al. (2024) does not reflect typical settings for NeSy predictors.

We argue both formally and empirically that the independence assumption is also a key limiting factor in such settings: For example, it is not possible for independent models to be RS-aware in the XOR MNIST problem of Example 1. They will either randomly get it correct, or find the RS in Fig. 1. In particular, we show that the conclusions of Faronius and Dos Martires (2025) are confounded by using a problem that does not contain RSs.

**Contributions.** We **C1)** provide a formalisation of the reasoning shortcut awareness of NeSy predictors in Section 4. We **C2)** prove that only in extremely rare cases, a NeSy predictor with the independence assumption can be RS-aware. Furthermore, we empirically highlight **C3)** that expressive models can be RS-aware (Section 3). However, this requires proper architecture and loss design decisions, which we discuss in Section 5.

## 2. Background: NeSy predictors and Reasoning Shortcuts

**Neurosymbolic Predictors.** We follow Marconato et al. (2023) and consider the discriminative problem of predicting a label $y \in \mathcal{Y}$ from a high-dimensional input $\mathbf{x} \in \mathcal{X}$. We assume access to a set of interpretable and discrete concepts $\mathcal{C}$ that, without loss of generality, we take to be the set of $k$-dimensional boolean vectors, i.e., $\mathcal{C} = \{0, 1\}^k$. Concepts can be understood as high-level, abstract representations of the input $\mathbf{x}$. However, we assume concepts are *not directly observed* in the data. Finally, we assume access to a program $\beta : \mathcal{C} \rightarrow \mathcal{Y}$ that maps concepts to labels. For a given label $y$, we define the constraint $\varphi_y(\boldsymbol{c}) = \mathbb{1}[\beta(\boldsymbol{c}) = y]$, which is 1 if and only if the program $\beta$ returns the label $y$ for concept $\boldsymbol{c}$. We use $\mathcal{C}_y = \{\boldsymbol{c} \in \mathcal{C} : \varphi_y(\boldsymbol{c}) = 1\}$ to denote the set of concepts consistent with $\varphi_y$.

NeSy predictors use a neural network to map from inputs $\mathbf{x}$ to concepts $\boldsymbol{c}$, and then use the program $\beta$ to predict the label $y$. One such class of models is the *(probabilistic) neurosymbolic (NeSy) predictors*. Probabilistic NeSy predictors use a concept distribution $p_{\boldsymbol{\theta}}(\boldsymbol{c} \mid \mathbf{x})$, often a neural network, to learn what concepts are likely to explain the input. Then, the data log-likelihood for a training pair $(\mathbf{x}, y)$ is given by these equivalent forms:

$$\log p_{\boldsymbol{\theta}}(y \mid \mathbf{x}) := \log \sum_{\boldsymbol{c} \in \mathcal{C}} p_{\boldsymbol{\theta}}(\boldsymbol{c} \mid \mathbf{x})\varphi_y(\boldsymbol{c}) = \log \sum_{\boldsymbol{c} \in \mathcal{C}_y} p_{\boldsymbol{\theta}}(\boldsymbol{c} \mid \mathbf{x}). \tag{1}$$

Equation 1 is analogous to (multi-instance) partial label learning (Wang et al., 2023) or disjunctive supervision (Zombori et al., 2024; Faronius and Dos Martires, 2025). NeSy predictors are evaluated along two axes. First, whether they find the input-label mapping, and second, whether they find the *ground-truth concept distribution $p^*(\boldsymbol{c} \mid \mathbf{x})$* defined below (Bortolotti et al., 2024).

**Independence assumption.** A common assumption taken in NeSy predictors is a conditional independence assumption in the model:

$$p_{\boldsymbol{\theta}}^{\perp\!\!\!\perp}(\boldsymbol{c} \mid \mathbf{x}) := \prod_{i=1}^{k} p_{\boldsymbol{\theta}}(c_i \mid \mathbf{x}) \tag{2}$$

Then, this model is used to compute the data log-likelihood in Equation 1. Implicitly, most practical NeSy predictors make this assumption (Xu et al., 2018; Badreddine et al., 2022; van Krieken et al., 2023). When $k > 1$, an independent concept distribution cannot represent all possible distributions over $\mathcal{C}$. Furthermore, given a fixed input $\mathbf{x}$ and label $y$, van Krieken et al. (2024) proved that the loss function of NeSy predictors 1) is usually highly non-convex, 2) has disconnected minima and 3) biases predictors towards overconfident hypotheses in the absence of evidence. All these problems are consequences of the limited expressivity of independent distributions.

Many probabilistic NeSy predictors (Xu et al., 2018; Manhaeve et al., 2018; Ahmed et al., 2022) realise Eq. 1 by performing efficient probabilistic reasoning over some compact logical representation (Darwiche and Marquis, 2002). The independence assumption is largely adopted because it greatly simplifies computation that is generally intractable[1] (Chavira and Darwiche, 2008). Commonly available neural probabilistic logic programming languages

---

1. We note that this assumption is not necessary to tractably compute Eq. 1 as shown in Vergari et al. (2021) and implemented in Ahmed et al. (2022), but this is only one tiny model in the vast NeSy literature sea.

like DeepProbLog (Manhaeve et al., 2018, 2021), NeurASP (Yang et al., 2020) and Scallop (Li et al., 2023) also assume independence over the symbols, or more precisely, over the probabilistic facts, in a program. Therefore, given a *fixed* program $\beta$ over a *fixed* number of probabilistic facts, there are many ground truth distributions involving dependencies that cannot be captured exactly, no matter the expressivity of the neural network. However, these languages are Turing complete probabilistic languages that can potentially represent any distribution (Taisuke, 1995; Poole and Wood, 2022; Faronius and Dos Martires, 2025). We resolve this apparent contradiction by noting that this is only possible by *augmenting* the program $\beta$ and introducing additional probabilistic facts. For example, we can add latent variables for a mixture model over independent distributions, or structured dependencies via Bayesian networks, thus going beyond the independence assumption as discussed in this paper. Nevertheless, in practice, and especially when empirically evaluating NeSy predictors over shared benchmarks that provide fixed programs $\beta$ (Bortolotti et al., 2024), this program augmentation is not present and Eq. 1 is realised with Eq. 2.

**Formal problem setup.** Next, we describe the formal assumptions behind how the data is generated, following Marconato et al. (2023).[2] In particular, we define the *ground-truth generative process* of the data as:

$$p^*(\boldsymbol{c}, \mathbf{x}, y) := p^*(\mathbf{x})p^*(\boldsymbol{c} \mid \mathbf{x})p^*(y \mid \boldsymbol{c}). \tag{3}$$

By marginalising, we obtain the ground-truth label distribution as:

$$p^*(y \mid \mathbf{x}) = \sum_{\boldsymbol{c} \in \mathcal{C}} p^*(\boldsymbol{c} \mid \mathbf{x})p^*(y \mid \boldsymbol{c}), \tag{4}$$

We consider two additional optional assumptions on the ground-truth process in Eq. 3, taken from Marconato et al. (2023):

- **Assumption A1**: Associated to each input in the support $\mathsf{supp}^*(\mathbf{x})$ of $p^*(\mathbf{x})$ is a unique, ground-truth concept $\mathbf{c}^*$. $\mathbf{c}^*$ is found with the oracle $f : \mathcal{C} \to \mathcal{Y}$ as $\mathbf{c}^* = f(\mathbf{x})$. Consequently, $p^*(\boldsymbol{c} \mid \mathbf{x})$ is a deterministic distribution.

- **Assumption A2**: Associated to each concept $\boldsymbol{c}$ is a unique label $y \in \mathcal{Y}$ found with the fixed program $\beta(\boldsymbol{c}) = y$. Consequently, $p^*(y \mid \boldsymbol{c})$ is a deterministic distribution.

Note that if both assumptions **A1** and **A2** hold, then $p^*(y \mid \mathbf{x})$ is a deterministic distribution where the label is given as $y = \beta(f(\mathbf{x}))$. If assumption **A2** holds, then we rewrite Eq. 4 as:

$$p^*(y \mid \mathbf{x}) = \sum_{\boldsymbol{c} \in \mathcal{C}} p^*(\boldsymbol{c} \mid \mathbf{x})\varphi_y(\boldsymbol{c}), \tag{5}$$

Considering Eq. 5, probabilistic NeSy predictors (Eq. 1) are a natural choice for this problem.

**Reasoning shortcuts.** A central issue in training NeSy predictors is detecting and mitigating *reasoning shortcuts*, which is when we learn the input-label mapping without properly learning the ground-truth concept distribution.

---

2. We use a simplified setup without vector-valued style variables $\mathbf{s}$ for notational simplicity.

**Definition 1** *Assume the ground-truth distribution $p^*$ where assumptions **A1** and **A2** hold. A function $\alpha : \mathcal{C} \to \mathcal{C}$ is a* (possible) concept remapping *if*

$$\forall \boldsymbol{c}^* \in \mathsf{supp}^*(\boldsymbol{c}) : \beta(\alpha(\boldsymbol{c}^*)) = \beta(\boldsymbol{c}^*), \tag{6}$$

*where $\mathsf{supp}^*(\boldsymbol{c})$ is the support of the distribution $p^*(\boldsymbol{c}) = \mathbb{E}_{p^*(\mathbf{x})}[p^*(\boldsymbol{c} \mid \mathbf{x})]$. If $\alpha$ is not the identity function, we say it is a* reasoning shortcut *(RS).[3] Associated to each concept remapping $\alpha$ is a* concept remapping distribution $p_\alpha(\boldsymbol{c} \mid \mathbf{x}) := \mathbb{1}[\alpha(f(\mathbf{x})) = \boldsymbol{c}]$.

Importantly, RSs $\alpha$ and their corresponding concept distribution $p_\alpha(\boldsymbol{c} \mid \mathbf{x})$ solve the input-label mapping. For each $\mathbf{x} \in \mathcal{X}$, $p_\alpha(y \mid \mathbf{x})$ deterministically returns $\beta(\alpha(f(\mathbf{x})))$. By Eq. 6, we have $\beta(\alpha(f(\mathbf{x}))) = \beta(f(\mathbf{x})) = y$, which is what the ground-truth label distribution $p^*(y \mid \mathbf{x})$ returns under **A1** and **A2**. However, the concept distribution is incorrect: non-identity mappings $\alpha$ will return a different concept $\boldsymbol{c}$ for some $\mathbf{x} \in \mathcal{X}$.

When training, we ideally converge on the ground-truth concept distribution $p^*(\boldsymbol{c} \mid \mathbf{x})$, and not an RS $p_\alpha(\boldsymbol{c} \mid \mathbf{x})$. Unfortunately, although techniques like specialised architectures or multitask learning can help, the most reliable method for finding $p^*$ is to provide costly supervision data directly on the concepts (Marconato et al., 2023). Instead, BEARS (Marconato et al., 2024) argues that for settings where we cannot disambiguate the ground-truth concept distribution from the RSs, we should be *aware* of the RSs. BEARS implements RS-awareness using an ensemble of NeSy predictors with the independence assumption to learn a collection of concept remappings. BEARS does *not* take the independence assumption (Eq. 2) as it mixes multiple conditionally independent models.

## 3. When is the independence assumption appropriate?

With the basic analysis tools under our belt, we discuss if the limited expressivity of the independence assumption is appropriate for NeSy predictors. When assumption **A1** does not hold, the independence assumption limits us: Assumption **A1** allows us to completely derive the ground-truth concepts $\boldsymbol{c}^*$ from the input $\mathbf{x}$. However, many settings violate assumption **A1**, such as under partial observability like self-driving cars, where part of the vision is blocked, or when planning under incomplete information. Then, the ground-truth concept distribution $p^*(\boldsymbol{c} \mid \mathbf{x})$ is not factorised, and NeSy predictors with the independence assumption cannot capture $p^*$.

When both assumptions **A1** and **A2** are satisfied, the ground-truth concept distribution $p^*(\boldsymbol{c} \mid \mathbf{x})$ is a deterministic mapping which can be *expressed* by conditionally independent distributions. So is checking for these two assumptions enough? No, as sufficient expressivity does not mean we actually *learn* the ground-truth concept distribution.

### 3.1. The independence assumption fails under reasoning shortcuts

Next, we provide a simple example where models using the independence assumption fail to learn a calibrated ground-truth concept distribution. In particular, we implement XOR MNIST from Example 1, which has a reasoning shortcut for independent models. We use

---

3. Our definition differs from Marconato et al. (2023) for readability and focus. We only consider deterministic RSs and true risk minimisers rather than log-likelihood minimisers.

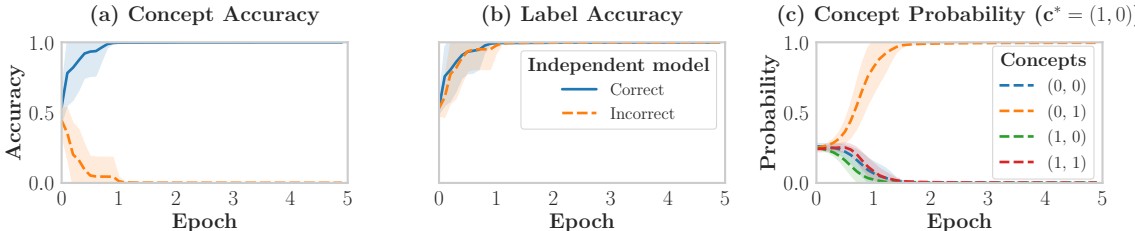

Figure 2: **Models with independence assumption are highly confident in the XOR reasoning shortcut.** Test concept and label accuracy during training for a conditionally independent model. We group the runs by whether they find the correct solution (solid line) or the reasoning shortcut (dashed line).

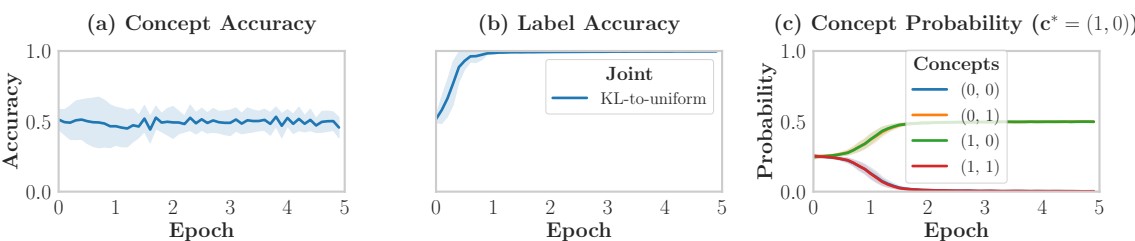

Figure 3: **Expressive models can solve the input-label problem while being RS-aware.** See Figure 2 for legend details.

the program $\beta(\boldsymbol{c}) = c_1 \wedge \neg c_2 \vee \neg c_1 \wedge c_2$ and the LeNet model for the concept distribution $p_{\boldsymbol{\theta}}^{\perp\!\!\!\perp}(\boldsymbol{c} \mid \mathbf{x})$. We fit the model by maximising Eq. 1. For more details, see Appendix C.

As shown in Fig. 2, all 20 runs solve the input-label mapping. However, 11 out of 20 runs learn the reasoning shortcut that maps all MNIST digits of 1 to 0, and MNIST digits of 0 to 1. Indeed, the theory of RSs gives only this remapping as an RS for this problem (Marconato et al., 2023)[4]. There is not enough information to disambiguate between the ground-truth concept distribution and the RS; both obtain perfect data-likelihoods.

How would we achieve an RS-aware predictor in this task for an independent model? Given any input, the marginal distribution of both digits should assign 0.5 probability to 1 and 0.5 probability to 0. Therefore, we assign 0.25 probability to each combination $\{(0,0),(0,1),(1,0),(1,1)\}$. But then we assign 0.5 probability to both labels $y = 1$ and $y = 0$, meaning we no longer solve the input-label mapping. Therefore, independent models cannot express the RS-aware concept distribution.

### 3.2. Can we train RS-aware expressive models?

If independent models are unable to represent uncertainty in the XOR MNIST task, then how about more expressive models? From an expressivity perspective, the answer is yes. But can we also *learn* a calibrated, reasoning shortcut aware model $p_{\boldsymbol{\theta}}(\boldsymbol{c} \mid \mathbf{x})$?

As shown in Faronius and Dos Martires (2025); Zombori et al. (2024), expressive models also exhibit a bias towards determinism during training. However, clever loss functions can circumvent this issue (Zombori et al., 2024). We minimise the KL-divergence to the uniform distribution over possible concepts. Given a pair $(\mathbf{x}, y)$ (Mendez-Lucero et al., 2024),

$$\mathrm{KL}\left[q(\boldsymbol{c} \mid y) \mid\mid p_{\boldsymbol{\theta}}(\boldsymbol{c} \mid \mathbf{x})\right] = -\frac{1}{|\mathcal{C}_y|} \sum_{\boldsymbol{c} \in \mathcal{C}_y} \log |\mathcal{C}_y| \cdot p_{\boldsymbol{\theta}}(\boldsymbol{c} \mid \mathbf{x}), \tag{7}$$

where $q(\boldsymbol{c} \mid y) := \frac{\varphi_y(\boldsymbol{c})}{|\mathcal{C}_y|}$ is the uniform distribution over concepts consistent with the label $y$. We adapt the LeNet model by horizontally joining the two MNIST images, and change the output layer to 4 classes corresponding to each concept combination.

In Fig. 3, we see that the model also learns the input-label mapping, but also learns to assign $0.5$ probability to the two valid concept combinations $(0, 1)$ and $(1, 0)$ when the input digits are different. Therefore, the model learned the RS-aware concept distribution. We emphasise that this is only possible because the model does not have the independence assumption. In fact, as we show in Appendix B, using Eq. 7 for the independence assumption fails to learn *anything* beyond the model's initialisation.

## 4. Models with independence cannot be reasoning shortcut aware

The above example motivates a closer study of how reasoning shortcuts are related to the independence assumption. In particular, we consider RS-awareness, the ability of a model to express uncertainty over multiple different RSs. We use the formal setup from Section 2 and take assumptions **A1** and **A2**. We assume a conditional constraint $\varphi_y$. Associated to each input $\mathbf{x}$ is a ground-truth concept $\boldsymbol{c}^* = f(\mathbf{x})$ and a unique label $y = \beta(\boldsymbol{c}^*)$. Recall that RSs are non-identity concept remappings $\alpha : \mathcal{C} \rightarrow \mathcal{C}$ which map ground-truth concepts $\boldsymbol{c}^*$ to potentially different concepts $\boldsymbol{c}$ using concept distributions $p_{\alpha}(\boldsymbol{c} \mid \mathbf{x}) := \mathbb{1}[\boldsymbol{c} = \alpha(f(\mathbf{x}))]$. We emphasise that $p_{\alpha}$ is unknown, as it requires the oracle $f$.

If the ground-truth concept distribution cannot be identified from the RSs, we want to be RS-aware by expressing uncertainty over multiple concept remappings. Proposition 3 of Marconato et al. (2023) shows *all* concept distributions $p(\boldsymbol{c} \mid \mathbf{x})$ solving the input-label mapping are convex combinations of a set of concept remapping distributions $p_{\alpha}$.

**Definition 2** *Let $\mathcal{A} = \{\alpha_1, ..., \alpha_m\}$ be a set of $m$ concept remappings and let $\boldsymbol{\pi} \in \Delta^{m-1}$, where $\Delta^{m-1}$ is the $(m-1)$-dimensional probability simplex, which is the set of all probability distributions over $m$ elements. We define a* reasoning shortcut mixture $p_{\mathcal{A}, \boldsymbol{\pi}}$ *as*

$$p_{\mathcal{A}, \boldsymbol{\pi}}(\boldsymbol{c} \mid \mathbf{x}) := \sum_{i=1}^{m} \pi_i p_{\alpha_i}(\boldsymbol{c} \mid \mathbf{x}). \tag{8}$$

---

4. This assumes independent models with *disentangled architectures*. For problems involving two MNIST digits, Marconato et al. (2023) understands this as architectures that use a single neural network for classifying both MNIST digits, thereby incorporating parameter sharing.

We next give a precise definition of RS-awareness, which we understand as the ability to represent these RS mixtures. We distinguish *complete mixing*, which is when all mixtures can be expressed, and *weak mixing*, which only requires representing a single mixture:

**Definition 3** *Let $\mathcal{A}$ be a set of concept remappings. We say a model class $p_{\boldsymbol{\theta}}(\boldsymbol{c} \mid \mathbf{x})$, $\boldsymbol{\theta} \in \Theta$ is* completely reasoning shortcut aware *over $\mathcal{A}$ if for all $\boldsymbol{\pi} \in \Delta^{k-1}$, there is a parameter $\boldsymbol{\theta}$ such that $p_{\boldsymbol{\theta}}(\boldsymbol{c} \mid \mathbf{x}) = p_{\mathcal{A},\boldsymbol{\pi}}(\boldsymbol{c} \mid \mathbf{x})$. Furthermore, a model class is* (weakly) reasoning shortcut aware *over $\mathcal{A}$ if it can represent $p_{\mathcal{A},\boldsymbol{\pi}}(\boldsymbol{c} \mid \mathbf{x})$ for some $\boldsymbol{\pi} \in \Delta^{k-1}$, where $0 < \pi_i < 1$ for all $i$.*

Our goal is to study under what conditions models with the independence assumption are RS-aware. Therefore, we must define precisely what we mean with this model class. For simplicity, we take the most powerful possible model class for the independence assumption. In realistic scenarios, we can only approximate this class.

**Definition 4** *The* universal conditionally independent (UCI) *model class is the set of all distributions formed from any function $\mu : \mathcal{X} \to [0,1]^k$ as*

$$p_{\mu}^{\perp\!\!\!\perp}(\boldsymbol{c} \mid \mathbf{x}) := \prod_{i=1}^{k} \mu(\mathbf{x})_i^{c_i} (1 - \mu(\mathbf{x})_i)^{1-c_i}. \tag{9}$$

All concept remapping distributions $p_\alpha$ fall within the UCI class by setting $\mu = \alpha(f(\mathbf{x}))$. How about RS mixtures $p_{\mathcal{A},\boldsymbol{\pi}}$? UCI models are only very rarely weakly RS-aware. To make this precise, we recall relevant definitions from van Krieken et al. (2024).

**Definition 5** *An* incomplete concept *$\boldsymbol{c}_D$ assigns values $\{0,1\}$ to a subset of the variables in $\boldsymbol{c}$ indexed by $D \subseteq [k]$. The* cover *$\mathcal{C}_{\boldsymbol{c}_D} \subseteq \mathcal{C}$ of $\boldsymbol{c}_D$ is the set of all concepts $\boldsymbol{c}$ that agree with $\boldsymbol{c}_D$ on the variables in $D$. Let $\varphi_y : \mathcal{C} \to \{0,1\}$ be the constraint induced by label $y$. An incomplete concept $\boldsymbol{c}_D$ is an* implicant *of $\varphi_y$ if and only if for all concepts $\boldsymbol{c}$ in the cover $\mathcal{C}_{\boldsymbol{c}_D}$, the constraint holds, that is, $\varphi_y(\boldsymbol{c}) = 1$.*

To formalise the link between implicants and RSs, we introduce *confusion sets*, the set of concepts that a ground truth concept $\boldsymbol{c}^*$ gets remapped into using concept remappings $\mathcal{A}$:

**Definition 6** *Given a set of concept remappings $\mathcal{A}$ and a ground-truth concept $\boldsymbol{c}^* \in \text{supp}^*(\boldsymbol{c})$, the* confusion set *$\mathcal{V}_{\boldsymbol{c}^*}$ is $\mathcal{V}_{\boldsymbol{c}^*} = \{\alpha(\boldsymbol{c}^*) \mid \alpha \in \mathcal{A}\} \subseteq \mathcal{C}_y$.*

We next obtain a necessary condition for the UCI model class to be (weakly) RS-aware over a set of RSs, and we provide some examples. All proofs are available in Appendix A.

**Theorem 7** *Given a set of concept remappings $\mathcal{A}$, if the UCI model class is (weakly) RS-aware over $\mathcal{A}$, then for all ground-truth concepts $\boldsymbol{c}^* \in \text{supp}^*(\boldsymbol{c})$, the confusion set $\mathcal{V}_{\boldsymbol{c}^*}$ is equal to the cover of an implicant $\boldsymbol{c}_D$ of $\varphi_y$, $y = \beta(\boldsymbol{c}^*)$.*

**Example 2** *XOR MNIST has the concept remappings $\alpha_1(\boldsymbol{c}) = \boldsymbol{c}$ (identity), and $\alpha_2(0,0) = (1,1)$, $\alpha_2(1,1) = (0,0)$, $\alpha_2(0,1) = (1,0)$, $\alpha_2(1,0) = (0,1)$ ($\alpha_2$ is the only RS for disentangled architectures, see Footnote 4). Therefore, the different confusion sets are:*

$$\mathcal{V}_{(0,0)} = \{(0,0),(1,1)\} \quad \mathcal{V}_{(1,0)} = \{(0,1),(1,0)\}$$
$$\mathcal{V}_{(0,1)} = \{(0,1),(1,0)\} \quad \mathcal{V}_{(1,1)} = \{(0,0),(1,1)\}$$

For $\varphi_0$, the implicant covers are $\{(0,0)\}$ and $\{(1,1)\}$. Note that neither of these covers are equal to the confusion sets $\mathcal{V}_{(0,0)}$ and $\mathcal{V}_{(1,1)}$. Similarly, for $\varphi_1$, the implicant covers are $\{(0,1)\}$ and $\{(1,0)\}$, none of which are equal to the confusion sets $\mathcal{V}_{(0,1)}$ and $\mathcal{V}_{(1,0)}$. Therefore, by Theorem 7, the UCI model class cannot be (weakly) RS-aware for the **XOR** MNIST problem.

What about a positive example? For $k = 2$, a problem with RSs that independent models can mix over is the following:

**Example 3** *Let $\beta(\boldsymbol{c}) = c_1$. Note that $\{(1,0),(1,1)\}$ and $\{(0,0),(0,1)\}$ are implicant covers of $\varphi_1$ and $\varphi_0$ respectively. The concept remappings are $\alpha_1(\boldsymbol{c}) = \boldsymbol{c}$ and $\alpha_2(\boldsymbol{c}) = (c_1, \neg c_2)$, as we have no evidence to disambiguate the value of $c_2$. The corresponding confusion sets therefore are:*

$$\mathcal{V}_{(0,0)} = \{(0,0),(0,1)\} \quad \mathcal{V}_{(1,0)} = \{(1,0),(1,1)\}$$
$$\mathcal{V}_{(0,1)} = \{(0,0),(0,1)\} \quad \mathcal{V}_{(1,1)} = \{(1,0),(1,1)\}$$

*Each of these are equal to the implicant covers of $\varphi_1$ and $\varphi_0$, meaning the necessary condition of Theorem 7 is satisfied. Given some probability $\pi$ of preferring $\alpha_1$ over $\alpha_2$, the UCI model class represents $p_{\mathcal{A},\boldsymbol{\pi}}(\boldsymbol{c} \mid \mathbf{x})$ with $\mu(\mathbf{x}) = (f(\mathbf{x})_1, \pi)$.*

This example may seem dull — The program $\beta$ gives perfect supervision for the first variable, and none for the second. But that is precisely the condition of Theorem 7. When the confusion set is equal to an implicant cover of $\boldsymbol{c}_D$, this means it agrees on $|D| \leq k$ of the variables. In other words, this is when we have *partial supervision* over some of the variables in $\boldsymbol{c}$. For practical NeSy setups, such programs are not meaningful.

Interestingly, for Example 3, UCI is not just weakly RS-aware, but also completely RS-aware. This requires a stronger condition, which is both necessary and sufficient:

**Theorem 8** *Let $\mathcal{A}$ be a set of concept remappings. Then the UCI model class is completely reasoning shortcut aware over $\mathcal{A}$ if and only if for all $\boldsymbol{c}^* \in \text{supp}^*(\boldsymbol{c})$, $\mathcal{V}_{\boldsymbol{c}^*}$ is a singleton, or $\mathcal{V}_{\boldsymbol{c}^*} = \{\boldsymbol{c}_1, \boldsymbol{c}_2\}$, where $\boldsymbol{c}_1$ and $\boldsymbol{c}_2$ differ in exactly one variable.*

Note that if a confusion set is a singleton, then that concept is not affected by any RSs as $\boldsymbol{c}^* \in \mathcal{V}_{\boldsymbol{c}^*}$. Therefore, UCI is only completely RS-aware if each concept is affected by at most one RS, and they differ in exactly one variable.

## 5. Expressiveness and architecture design

If expressive models allow us to be RS-aware, should we use them for any task? Faronius and Dos Martires (2025) showed that expressive models fail to learn the correct concept distribution for a problem *without* RSs. In particular, they use the **Traffic Lights** MNIST problem, which is like **XOR** MNIST (Example 1) except using the program $\beta(\boldsymbol{c}) = \neg c_1 \vee \neg c_2$.

Why does this problem not contain RSs for independent models? The constraint $\varphi_0 = c_1 \wedge c_2$ provides completely supervised information: the only option is $c_1 = c_2 = 1$. Therefore, given enough data, the model can learn to correctly classify MNIST digits of 1, leaving only the problem of classifying MNIST digits of 0. But now, given input digits that are different, the model can learn to recognise which of the pairs $(0,1)$ is not a 1, and then learn to also

Table 1: **Well-designed expressive models can be RS-aware while solving problems without RSs.** We compare an independent model with two expressive models using different losses on a problem with (XOR MNIST) and without (Traffic Lights MNIST) reasoning shortcuts. We report label accuracy, concept accuracy and concept calibration with expected calibration error. The statistically best model-loss combinations are bolded. For details on experimental setup, see Appendix C.

| Method | Traffic Lights MNIST (No RS) | | | XOR MNIST (Has RS) | | |
|---|---|---|---|---|---|---|
| | $\text{Acc}_\mathbf{y}\uparrow$ | $\text{Acc}_c\uparrow$ | $\text{ECE}_c\downarrow$ | $\text{Acc}_\mathbf{y}\uparrow$ | $\text{Acc}_c\uparrow$ | $\text{ECE}_c\downarrow$ |
| Independent - Semantic Loss | **99.91± 0.07** | 99.87± 0.04 | 0.22± 0.06 | 99.83± 0.07 | **45.01± 50.96** | **54.96± 49.55** |
| Independent - Uniform-KL | 70.58± 0.00 | 46.32± 0.02 | 30.17± 0.33 | 51.47± 0.00 | 46.31± 0.00 | 28.69± 0.01 |
| Joint - Semantic Loss | 99.86± 0.05 | 69.32± 5.73 | 29.74± 6.10 | 99.73± 0.13 | **53.55± 15.04** | 45.03± 15.47 |
| Joint - Uniform-KL | 99.86± 0.05 | 75.73± 0.03 | 0.77± 0.09 | 99.73± 0.14 | 45.61± 7.22 | **6.62± 5.78** |
| AR - Semantic Loss | **99.91± 0.09** | **99.91± 0.04** | **0.09± 0.04** | **99.86± 0.07** | 45.00± 50.97 | 54.97± 49.66 |
| AR - Uniform-KL | 99.83± 0.13 | 88.83± 0.10 | 14.98± 1.52 | **99.88± 0.06** | **50.99± 12.61** | 10.24± 7.06 |

classify 0s. In Table 1, we compare the performance of the independent model with two expressive models. Like in Faronius and Dos Martires (2025), the independent model obtains high concept accuracy, while the joint model from Section 3 obtains high label accuracy but poor concept accuracy. This is because the joint model cannot distinguish between the concept configurations $(0,0)$, $(0,1)$ and $(1,0)$: these are simply different output logits.

However, we find that a well-designed autoregressive model can solve Traffic Lights MNIST with high concept accuracy, while also being RS-aware on XOR MNIST under Eq. 7, showcased by low expected calibration error under high label accuracy. This highlights that it is not as simple as just using any expressive model for NeSy predictors — we also need to use neural network architectures with the right inductive biases for the problem at hand.

## 6. Conclusion

We studied under what conditions taking the independence assumption is appropriate for NeSy predictors. We proved that models with the independence assumption can only be aware of reasoning shortcuts in very limited settings. This limits the reliability of NeSy predictors with independence, meaning they might silently fail out-of-distribution. Finally, we showed that expressive models that go beyond the independence assumption can be RS-aware. However, we also found that neural network design is a significant factor.

Given these results, studying what expressive architecture design suits NeSy predictors is a fruitful direction for future research. Existing work used mixtures of independent models (Marconato et al., 2024), which is especially powerful when using probabilistic circuits (Ahmed et al., 2022) and discrete diffusion models (van Krieken et al., 2025), which showed significant improvements over independent models on the RSBench dataset (Bortolotti et al., 2024). Furthermore, we believe studying benchmarks that go beyond the naive assumption of full observability (Assumption **A1** in Section 2) would help elicit differences between the behaviour of NeSy predictors with independence and those without.

## Acknowledgements

Emile van Krieken was funded by ELIAI (The Edinburgh Laboratory for Integrated Artificial Intelligence), EPSRC (grant no. EP/W002876/1). Pasquale Minervini was partially funded by ELIAI, EPSRC (grant no. EP/W002876/1), an industry grant from Cisco, and a donation from Accenture LLP. Antonio Vergari was supported by the "UNREAL: Unified Reasoning Layer for Trustworthy ML" project (EP/Y023838/1) selected by the ERC and funded by UKRI EPSRC. Finally, we would like to express our gratitude to Emanuele Marconato, Pedro Zuidberg Dos Martires and the anonymous reviewers for useful feedback and discussions during the writing of this paper.

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

## Appendix A. Proofs

We start by introducing concept remapping distributions, which give the probability that an RS mixture (Theorem 2) remaps $\boldsymbol{c}^*$ to $\boldsymbol{c}$.

**Definition 9** *Given a set of deterministic RSs $\mathcal{A}$ and some weight $\boldsymbol{\pi} \in \Delta^m$, we define the RS mixture remapping distribution $p_{\mathcal{A},\boldsymbol{\pi}}(\boldsymbol{c} \mid \boldsymbol{c}^*)$ as*

$$p_{\mathcal{A},\boldsymbol{\pi}}(\boldsymbol{c} \mid \boldsymbol{c}^*) := \sum_{i=1}^{m} \pi_i \mathbb{1}[\boldsymbol{c} = \alpha_i(\boldsymbol{c}^*)]. \tag{10}$$

Note that the support of $p_{\mathcal{A},\boldsymbol{\pi}}(\boldsymbol{c} \mid \mathbf{x})$ is equal to $\mathcal{V}_{\boldsymbol{c}^*}$ if $0 < \pi_i < 1$ for all $i$. Furthermore, as a simple base case where there is only a simple concept remapping $\mathcal{A} = \{\alpha\}$, then $p_{\{\alpha\},1}(\boldsymbol{c} \mid \boldsymbol{c}^*) = \mathbb{1}[\boldsymbol{c} = \alpha(\boldsymbol{c}^*)]$ is a deterministic distribution.

**Lemma 10** *The UCI model class can represent the RS mixture distribution $p_{\mathcal{A},\boldsymbol{\pi}}(\boldsymbol{c} \mid \mathbf{x})$ if and only if $p_{\mathcal{A},\boldsymbol{c}}(\boldsymbol{c} \mid \boldsymbol{c}^*)$ is a factorised distribution for all $\boldsymbol{c}^* \in supp^*(\boldsymbol{c})$.*

**Proof** $\implies$ Assume the UCI model class represents $p_{\mathcal{A},\boldsymbol{\pi}}(\boldsymbol{c} \mid \mathbf{x})$ with function $\mu$. Then for all $\mathbf{x} \in \text{supp}^*(\mathbf{x})$,

$$p_{\mu}^{\perp\!\!\!\perp}(\boldsymbol{c} \mid \mathbf{x}) = p_{\mathcal{A},\boldsymbol{\pi}}(\boldsymbol{c} \mid \mathbf{x}) = \sum_{i=1}^{m} \pi_i p_{\alpha_i}(\boldsymbol{c} \mid \mathbf{x}) = \sum_{i=1}^{m} \pi_i \mathbb{1}[\boldsymbol{c} = \alpha(f(\mathbf{x}))].$$

Letting $\boldsymbol{c}^* = f(\mathbf{x})$, which must be in $\mathrm{supp}^*(\boldsymbol{c})$, we have

$$p_\mu^{\perp\!\!\!\perp}(\boldsymbol{c} \mid \mathbf{x}) = \sum_{i=1}^m \pi_i \mathbb{1}[\boldsymbol{c} = \alpha(\boldsymbol{c}^*)] = p_{\mathcal{A},\boldsymbol{\pi}}(\boldsymbol{c} \mid \boldsymbol{c}^*) \tag{11}$$

Since $p_\mu^{\perp\!\!\!\perp}(\boldsymbol{c} \mid \mathbf{x})$ is a factorised distribution, $p_{\mathcal{A},\boldsymbol{\pi}}(\boldsymbol{c} \mid \boldsymbol{c}^*)$ must be too.

$\impliedby$ Assume $p_{\mathcal{A},\boldsymbol{c}}(\boldsymbol{c} \mid \boldsymbol{c}^*)$ is factorised. Then, for each $\mathbf{x} \in \mathrm{supp}(\mathbf{x})$, let $\boldsymbol{c}^* = f(\mathbf{x})$,

$$p_{\mathcal{A},\boldsymbol{\pi}}(\boldsymbol{c} \mid \mathbf{x}) = \sum_{i=1}^m \pi_i \mathbb{1}[\boldsymbol{c} = \alpha(f(\mathbf{x}))] = \sum_{i=1}^m \pi_i \mathbb{1}[\boldsymbol{c} = \alpha(\boldsymbol{c}^*)] = p_{\mathcal{A},\boldsymbol{\pi}}(\boldsymbol{c} \mid \boldsymbol{c}^*), \tag{12}$$

and hence $p_{\mathcal{A},\boldsymbol{\pi}}(\boldsymbol{c} \mid \mathbf{x})$ is also factorised. Therefore, it can be represented by the product of its marginals.

In particular, we define the function $\mu$ to equal the vector of marginals of $p_{\mathcal{A},\boldsymbol{\pi}}(\boldsymbol{c} \mid \mathbf{x})$:

$$\mu(\mathbf{x})_j = p_{\mathcal{A},\boldsymbol{\pi}}(c_i = 1 \mid \mathbf{x}) = \sum_{i=1}^m \pi_i \mathbb{1}[\alpha_i(f(\boldsymbol{c}))_j = 1], \tag{13}$$

which constructs a distribution $p_\mu^{\perp\!\!\!\perp}(\boldsymbol{c} \mid \mathbf{x})$ in UCI. ∎

**Theorem 11** *Given a set of concept remappings $\mathcal{A}$, if the UCI model class is (weakly) RS-aware over $\mathcal{A}$, then for all ground-truth concepts $\boldsymbol{c}^* \in \mathrm{supp}^*(\boldsymbol{c})$, the confusion set $\mathcal{V}_{\boldsymbol{c}^*}$ is equal to the cover of an implicant $\boldsymbol{c}_D$ of $\varphi_y$, $y = \beta(\boldsymbol{c}^*)$.*

**Proof** First, assume the UCI model class mixes over $\mathcal{A}$. We assume this is done for weight $\boldsymbol{\pi}$ using function $\mu$. That is, using Eq. 12, $p_\mu(\boldsymbol{c} \mid \mathbf{x}) = p_{\mathcal{A},\boldsymbol{\pi}}(\boldsymbol{c} \mid \mathbf{x}) = p_{\mathcal{A},\boldsymbol{\pi}}(\boldsymbol{c} \mid \boldsymbol{c}^* = f(\mathbf{x}))$ for all $\mathbf{x} \in \mathrm{supp}^*(\mathbf{x})$. Therefore, the support of $p_\mu(\boldsymbol{c} \mid \mathbf{x})$ is equal to that of $p_{\mathcal{A},\boldsymbol{\pi}}(\boldsymbol{c} \mid \mathbf{x})$, which is $\mathcal{V}_{\boldsymbol{c}^*}$.

By condition 6, we have that for all $\boldsymbol{c} \in \mathcal{V}_{\boldsymbol{c}^*}$, $\beta(\boldsymbol{c}) = \beta(\boldsymbol{c}^*) = y$. Therefore, $p_\mu(\boldsymbol{c} \mid \mathbf{x})$ is a possible independent distribution for $\varphi_y$. That is, for all $\boldsymbol{c} \in \mathcal{C}$ such that $p_\mu(\boldsymbol{c} \mid \mathbf{x}) > 0$, we have that $\varphi_y(\boldsymbol{c}) = 1$. Then by Theorem 4.3 from van Krieken et al. (2024), we have that the deterministic component $\boldsymbol{c}_D$ of $p_\mu(\boldsymbol{c} \mid \mathbf{x})$ is an implicant of $\varphi_y$. Hence, all concepts in $\mathcal{V}_{\boldsymbol{c}^*}$ extend $\boldsymbol{c}_D$. Furthermore, the support of $p_\mu(\boldsymbol{c} \mid \mathbf{x})$ is equal to the cover of $\boldsymbol{c}_D$ (see proof of Theorem 4.3 in van Krieken et al. (2024)). Hence, $\mathcal{V}_{\boldsymbol{c}^*} = \mathcal{C}_{\boldsymbol{c}_D}$. ∎

**Theorem 12** *Let $\mathcal{A}$ be a set of concept remappings. Then the UCI model class is completely reasoning shortcut aware over $\mathcal{A}$ if and only if for all $\boldsymbol{c}^* \in \mathrm{supp}^*(\boldsymbol{c})$, $\mathcal{V}_{\boldsymbol{c}^*}$ is a singleton, or $\mathcal{V}_{\boldsymbol{c}^*} = \{\boldsymbol{c}_1, \boldsymbol{c}_2\}$, where $\boldsymbol{c}_1$ and $\boldsymbol{c}_2$ differ in exactly one variable.*

**Proof** $\implies$ Assume that the UCI model class is completely RS-aware over $\mathcal{A}$. By Theorem 7, we have that, for all $\boldsymbol{c}^* \in \mathrm{supp}^*(\boldsymbol{c})$, $\mathcal{V}_{\boldsymbol{c}^*}$ is equal to the cover of an implicant $\boldsymbol{c}_D$ of $\varphi_y$, where $y = \beta(\boldsymbol{c}^*)$. Furthermore, by the definition of $\mathcal{V}_{\boldsymbol{c}^*}$, we have that $|\mathcal{A}| \geq |\mathcal{V}_{\boldsymbol{c}^*}|$.

Consider some subset $\mathcal{A}' \subseteq \mathcal{A}$ of size $|\mathcal{V}_{\boldsymbol{c}^*}|$ for which the confusion set $\mathcal{V}'_{\boldsymbol{c}^*} = \mathcal{V}_{\boldsymbol{c}^*}$ is the same as for $\mathcal{A}$. We will next construct a weight vector $\boldsymbol{\pi}$ for $\mathcal{A}$. For all $\alpha_i \notin \mathcal{A}'$, we assign

$\pi_i = 0$. Now, we have a unique concept remapping $\alpha \in \mathcal{A}'$ such that $\alpha(\boldsymbol{c}^*) = \boldsymbol{c}$. Identify this $\alpha$ as $\alpha_{\boldsymbol{c}}$ and corresponding weight as $\pi_{\boldsymbol{c}}$. Then,

$$p_{\mathcal{A},\boldsymbol{\pi}}(\boldsymbol{c} \mid \boldsymbol{c}^*) = \pi_{\boldsymbol{c}} \mathbb{1}[\boldsymbol{c} \in \mathcal{V}_{\boldsymbol{c}^*}]$$

That is, to be completely RS-aware necessitates representing any distribution over $\mathcal{V}_{\boldsymbol{c}^*}$. By the fact that $\mathcal{V}_{\boldsymbol{c}^*}$ is the cover of an implicant, it is the set of all concepts with $\log_2(|\mathcal{V}_{\boldsymbol{c}^*}|)$ free variables. Independent distributions can only represent distributions over 0 or 1 free variables, and so we require $|\mathcal{V}_{\boldsymbol{c}^*}| \leq 2$.

If there are no free variables, we have the singleton $\mathcal{C}_{\boldsymbol{c}^*} = \{\boldsymbol{c}_1\}$. If there is one free variable, then by the fact that $\mathcal{C}_{\boldsymbol{c}^*}$ is the cover of an implicant, we have the two-element set $\mathcal{C}_{\boldsymbol{c}^*} = \{\boldsymbol{c}_1, \boldsymbol{c}_2\}$ that differ in some dimension $i$.

$\impliedby$ First, we show that for all $\boldsymbol{c}^* \in \mathrm{supp}^*(\boldsymbol{c})$, $\mathcal{V}_{\boldsymbol{c}^*}$ is a cover of an implicant. In the singleton case, the implicant is the complete concept $\boldsymbol{c}_1$, and in the case with two concepts $\boldsymbol{c}_1$ and $\boldsymbol{c}_2$, the implicant is the incomplete concept consisting of all variables except the one where $\boldsymbol{c}_1$ and $\boldsymbol{c}_2$ differ, namely $i$. The cover of this implicant consists of $\boldsymbol{c}_1$ and $\boldsymbol{c}_2$.

Let $\boldsymbol{\pi}$ be a weight vector for $\mathcal{A}$. Then, for each $\mathbf{x} \in \mathrm{supp}^*(\mathbf{x})$ and letting $\boldsymbol{c}^* = f(\mathbf{x})$, we define

$$\mu(\mathbf{x})_j = \begin{cases} c_{1,j} & \text{if } \mathcal{V}_{\boldsymbol{c}^*} = \{\boldsymbol{c}_1\} \\ c_{1,j} & \text{if } \mathcal{V}_{\boldsymbol{c}^*} = \{\boldsymbol{c}_1, \boldsymbol{c}_2\} \text{ and } c_{1,j} = c_{2,j} \\ \sum_{i=1}^m \pi_i \alpha_i(\boldsymbol{c}^*)_j & \text{if } \mathcal{V}_{\boldsymbol{c}^*} = \{\boldsymbol{c}_1, \boldsymbol{c}_2\} \text{ and } c_{1,j} \neq c_{2,j}. \end{cases} \tag{14}$$

Then we show that $p_\mu(\boldsymbol{c} \mid \mathbf{x})$ is equal to $p_{\mathcal{A},\boldsymbol{\pi}}(\boldsymbol{c} \mid \mathbf{x})$. Using Equation 11,

$$p_{\mathcal{A},\boldsymbol{\pi}}(\boldsymbol{c} \mid \mathbf{x}) = \sum_{i=1}^m \pi_i \mathbb{1}[\boldsymbol{c} = \alpha_i(\boldsymbol{c}^*)]$$

First, assuming $\mathcal{V}_{\boldsymbol{c}^*} = \{\boldsymbol{c}_1\}$, we have that

$$p_{\mathcal{A},\boldsymbol{\pi}}(\boldsymbol{c} \mid \mathbf{x}) = \mathbb{1}[\boldsymbol{c} = \boldsymbol{c}_1] \sum_{i=1}^m \pi_i = \mathbb{1}[\boldsymbol{c} = \boldsymbol{c}_1] = p_\mu(\boldsymbol{c} \mid \mathbf{x}).$$

Next, assuming $\mathcal{V}_{\boldsymbol{c}^*} = \{\boldsymbol{c}_1, \boldsymbol{c}_2\}$, then $\boldsymbol{c}_1$ and $\boldsymbol{c}_2$ differ only on variable $j$, one being 1 and the other 0. Then we have

$$p_{\mathcal{A},\boldsymbol{\pi}}(\boldsymbol{c} \mid \mathbf{x}) = \mathbb{1}[\boldsymbol{c}_{\backslash j} = \boldsymbol{c}_{1,\backslash j}] \sum_{i=1}^m \pi_i \mathbb{1}[c_j = \alpha_i(\boldsymbol{c}^*)_j]$$

$$= \mathbb{1}[\boldsymbol{c}_{\backslash j} = \boldsymbol{c}_{1,\backslash j}] \left( \sum_{i=1}^m \pi_i \alpha_i(\boldsymbol{c}^*)_j \right)^{c_j} \left( 1 - \sum_{i=1}^m \pi_i \alpha_i(\boldsymbol{c}^*)_j \right)^{1-c_j}$$

$$= \mathbb{1}[\boldsymbol{c}_{\backslash j} = \boldsymbol{c}_{1,\backslash j}] \mu(\mathbf{x})_j^{c_j} (1 - \mu(\mathbf{x})_j)^{1-c_j} = p_\mu(\boldsymbol{c} \mid \mathbf{x}).$$

$\blacksquare$

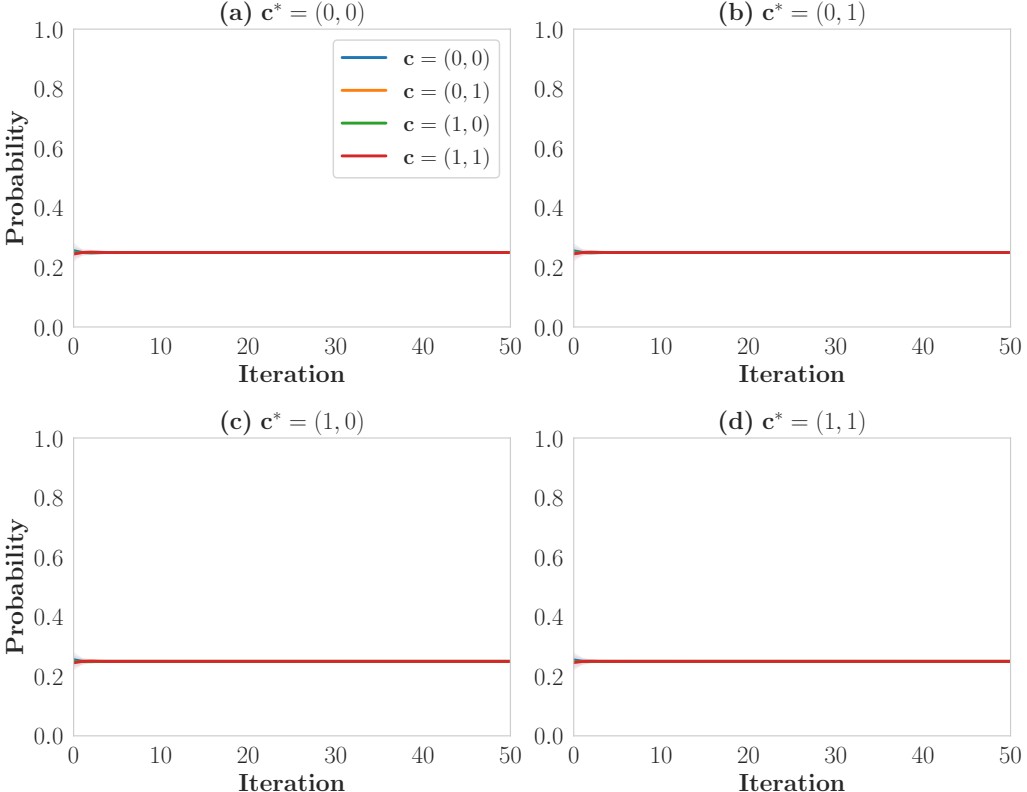

Figure 4: **An independent distribution fails to learn anything under RS-aware loss functions.** The evolution of test data probability predictions over the four possible concepts by minimising the KL-divergence in Equation 7 for a conditionally independent model.

## Appendix B. Minimising the uniform-KL for independent models

As we show in Fig. 4, using the KL-divergence and neural network architecture of Section 3.2 adapted to the independence assumption, we fail to learn *anything* beyond the model's initialisation. This might seem like a bug, but it is not: under independence, the KL-divergence is minimised in the parity problem when the probability of $c_1$ and $c_2$ are both 0.5, regardless of the input and label.[5]

## Appendix C. Experimental details

**Dataset construction.** We take the MNIST dataset, filter by the digits 0 and 1, and then construct pairs of digits by creating a random permutation of the remaining digits. This ensures each digit only appears once in each pair, and there is no test-data leakage. We used

---

5. For the case $y = 0$ in which the MNIST digits are the same, the KL-divergence under independence is equal to $\frac{1}{2}\left(-\log 2\mu_{c_1}\mu_{c_2} - \log 2(1 - \mu_{c_1})(1 - \mu_{c_2})\right)$, which is a convex function with unique minimiser at $\mu_{c_1} = \mu_{c_2} = 0.5$.

the standard MNIST train-test split for creating the test set. Then, we find the label using the programs $\beta$ for both the XOR MNIST and Traffic Lights MNIST problems.

**Training details.** Our code is available at https://github.com/HEmile/independence-vs-rs/tree/main. All experiments are run for 5 epochs on the training data. We did not do any hyperparameter tuning: all experiments are run with a learning rate of 0.0001 with Adam and a batch size of 64.

**Statistics.** All results are averaged over 20 random seeds, and report the mean and standard deviation of the results. In Table 1, we bold results that are statistically indistinguishable at $p = 0.05$ from the best performing model-loss combination using an unpaired Mann-Whitney U test. We highlight that several results, especially for XOR MNIST, are highly bi-modal, meaning sometimes even at 20 runs there is no statistically significant difference despite a large difference in averages.

**Model architectures.** We use the standard LeNet CNN architecture as the base for all our models in both XOR MNIST and Traffic Lights MNIST. This architecture contains a CNN encoder and an MLP classifier. More precisely, this uses 2 convolutional layers with maxpooling and ReLU activations which are concatenated into a 256-dimensional embedding. Then, we use a hidden layer to a 120-dimensional embedding and to an 84-dimensional embedding, each with ReLU activations. Finally, we use a sigmoid output layer which outputs a single logit activated by the sigmoid function.

In particular,

- **Independent model:** An independent model is given as $p_{\boldsymbol{\theta}}^{\perp\!\!\!\perp}(\boldsymbol{c} \mid \mathbf{x}) := p_{\boldsymbol{\theta}}(c_1 \mid \mathbf{x}) \cdot p_{\boldsymbol{\theta}}(c_2 \mid \mathbf{x})$. We use a single LeNet model for both digits, that is, $p_{\boldsymbol{\theta}}(c_1 \mid \mathbf{x})$ and $p_{\boldsymbol{\theta}}(c_2 \mid \mathbf{x})$ are the same model. Therefore, the independent model is disentangled in the sense of Footnote 4. We adopt the architecture to give a single logit as output, and apply a sigmoid to get the probability for the digit being 1, and 1 minus that for the digit being 0.

- **Joint model:** The joint model directly outputs a distribution over all 4 concept configurations using a model $p_{\boldsymbol{\theta}}(\boldsymbol{c} \mid \mathbf{x})$. We adapt the LeNet architecture as follows: We use a shared CNN encoder for both digits to encode the MNIST images. Then, we concatenate the two resulting embeddings, and feed them through the classifier. The classifier uses 4 outputs, for each combination of the two digits $(0,0), (0,1), (1,0), (1,1)$.

- **Autoregressive model:** Autoregressive models are defined as $p_{\boldsymbol{\theta}}^{AR}(\boldsymbol{c} \mid \mathbf{x}) := p_{\boldsymbol{\theta}}(c_1 \mid \mathbf{x}) \cdot p_{\boldsymbol{\theta}}(c_2 \mid \mathbf{x}, c_1)$. Similar to the joint model, we use a shared CNN encoder for both digits to encode the MNIST images. We implement the terms $p_{\boldsymbol{\theta}}(c_1 \mid \mathbf{x})$ and $p_{\boldsymbol{\theta}}(c_2 \mid \mathbf{x}, c_1)$ with a shared classifier that, like the independent model, outputs a single logit and applies a sigmoid. However, we add four extra inputs to the classifier. For the term $p_{\boldsymbol{\theta}}(c_1 \mid \mathbf{x})$, the first of these four inputs is always 1, and the rest are 0. This indicates that the classifier is dealing with the left digit. For the term $p_{\boldsymbol{\theta}}(c_2 \mid \mathbf{x}, c_1)$, the second input is 1 if $c_1 = 0$, while the third input is 1 if $c_1 = 1$. Furthermore, its fourth input is a 1-dimensional embedding of the left digit using a simple linear layer. This allows the classifier to condition on the contents of the left digit.

We note that the design of the autoregressive model is quite specific. We also tried different designs that simply concatenated the two digits, like for the joint model. However, we found

that when doing this for Traffic Lights MNIST, the autoregressive model would become overly confident in a single valid concept for the class $y = 1$, just like the joint model. By restricting the classifier of the second digit to only a 1-dimensional embedding of the left digit, we prevent the model from overfitting to a single concept.

**Expected calibration error.** The expected calibration error (ECE) (Guo et al., 2017) is a measure for assessing how well a probabilistic classifier knows what it does not know. It understands the probability given by the probabilistic classifier as predicting the ratio of how often it would be wrong on unseen data. For instance, for all test data points on which the classifier predicts (around) 0.5 probability, it is indeed right on half of the cases, and wrong on the other half. Similarly, if the classifier is confident, e.g. close to 1 probability for the positive class, it should also rarely be a negative class. We used the existing implementation from (Marconato et al., 2024).

The ECE formalises this by binning predictions using $K$ equally spaced probability bins. In particular, let $\text{Acc}(\mathcal{D}, f)$ measure the accuracy of the classifier $f$ on binary classification dataset $\mathcal{D}$, and let $\text{prob}(\mathcal{D}, f)$ be the average predicted probability of $f$ on the dataset. Then the ECE is defined as

$$\text{ECE}(\mathcal{D}, f) := \sum_{i=1}^{K} \frac{|\mathcal{D}_i|}{|\mathcal{D}|} |\text{Acc}(\mathcal{D}_i, f) - \text{Conf}(\mathcal{D}_i, f)| \tag{15}$$

where $\mathcal{D}_i$ is the subset of datapoints $(\mathbf{x}, \boldsymbol{c}) \in \mathcal{D}$ on which a probability between $\frac{i}{K}$ and $\frac{i+1}{K}$ is predicted by the classifier $f$. In our evaluation, we micro-average over the left and the right digits by simply concatenating the datasets.

