# OpenReview forum: "Neurosymbolic Reasoning Shortcuts under the Independence Assumption"
_nesyconf.org/NeSy/2025/Conference_Phase_2 — NeSy 2025 - Phase 2 Poster_

### Official Review · Reviewer_nuFR · 2025-06-30

**Rating:** 7
**Confidence:** 5

**Review:**

# Summary
The authors investigate the implications of the independence assumptions on the ability of
NeuroSymbolic models to represent uncertainty in learning latent concepts under weak supervisions.
This is particularly important to convey awareness of reasoning shortcuts, pathological behaviour preventing
the model from acquiring the right semantics for the unsupervised concepts. Theoretical findings demonstrate
that the independence assumption has serious shortcomings in representing uncertainty, which is captured
only under specific assumptions holding for the problem at hand.

# Soundness
The paper is technically sound and well-grounded. The proofs for the claims are provided in the Appendix
and the results are clearly demonstrated and justified. The experimental part lacks some details
and can be improved (see Weaknesses).

# Presentation
The paper is well written and easy to follow. It builds on the existing litterature on
NeuroSymbolic and Reasoning Shortcuts and clearly presents its contribution. Overall I am satisfied
with the presentation of the article.

# Contribution
The authors offer theoretical insights into the expressive capabilities of NeuroSymbolic models with and
without independence assumption. For the former case, the results are illustrated through Theorem 7 and 8.
For the latter, the article relies on proof of concepts, which is effective in conveying the idea.
Additionally, the framework introduced through Definitions 2 to 6 provides useful foundation for future research on this topic.

# Strenghts
The strenghts of the article is in its theoretical contributions, namely:

1. The formalization of reasoning shortcut awareness in Section 4 is valuable and very interesting in my opinion;
2. Theorems 7 and 8 provide useful insights into reasoning shortcut awareness for independent models.

Also, the empirical proofs of concept on how expressive models differ from the independent once are convincing enough.

# Weaknesses
As mentioned before, the experimental part could benefit further elaboration. In particular,
contribute C4 says: "*we discuss architecture and loss design decisions for NeSy predictors,
both in the presence of RSs and without, in Section 5*" However Section 5 is not detailed enough
to fully justify this contribution. Indeed, while it gives some insights, it focuses on a single model
(LeNet) and two loss functions (Semantic Loss and Uniform-KL). This is not necessarily a problem,
but I would merge contributes C3 and C4 as a whole, since they are both empirical proof of concepts.
While LeNet is suitable for the proposed tasks, a comparison with an additional architecture would
strengthen the findings. Also, adding other losses (e.g. LTN) could benefit the analysis.
Finally, extending the evaluation to a non-MNIST task would add valuable depth to the experimental setting.



# Other Comments
- The article mentions '*disentanglement*' a few times. It would be helpful to define it (e.g. in Section 2);
- The Appendix C is missing the architecture for the MLP used and the specification of technical environment setup;
- In Table 1 the Expected Calibration Error (ECE) on the concepts is not defined. While many readers are familiar with
the metric already, a brief inline definition would improve clarity;
- Definition 1: $f^{-1}(c)$ should be $f^{-1}(x)$;
- Example 2: $\varphi_1$ and $\varphi_0$ appear to be swapped.


# Question
Can you please confirm whether the independent models is Section 5 are disentangled?

# Evaluation
I recommed acceptance, as the theoretical findings and the proposed framework for uncertainty modelling
are valuable and interesting on their own. While the experimental part could be improved,
the current proof-of-concept experiments are sufficient to support the paper’s claims.

**Anonymity:**

Remain anonymous

---

### Official Review · Reviewer_rtsF · 2025-07-06
**Interesting results**

**Rating:** 8
**Confidence:** 4

**Review:**

The paper's aim is to show formally showing that assuming independence among symbolic concepts entails that a model can never represent uncertainty over certain concept combinations.

Overall, ther paper is well written and trackles an interesting problem. It is a bit strong to say that one paper (Faronius and Dos Martires, 2025) shows that "there is, however, scepticism in the NeSy community about the scenarios" in a social not just existential logically sense. Mroeover, the paper should also discuss obvious alternatives to the independency assumption like probabilsitic circuits or some chain-rule probablistic model (that can even be compressed via a neural network), as discussed on page implicitely but kind of dismissed by arguing that this is not used in practice. The point here is to dicuss the significance of the presentd result. This discussion should also consider how likely the worst case scenario is met in reality as well as to which extend the result is obvious and indeep unavoidable. The experiment of 3.2 is going into the direction, thanks, but still it is a bit like saying that Naive Bayes is not a ganeral density estimator, which is true but also potentially obvious.

Nevertheless, I really like the paper. It is well written and will help the NeSy community.

**Anonymity:**

Remain anonymous